# Characteristics of Cancer Stem Cells and Their Potential Role in Endometrial Cancer

**DOI:** 10.3390/cancers16061083

**Published:** 2024-03-07

**Authors:** Karolina Frąszczak, Bartłomiej Barczyński

**Affiliations:** 11st Chair and Department of Oncological Gynaecology and Gynaecology, Medical University in Lublin, 20-081 Lublin, Poland; karolina.fraszczak@umlub.pl; 2Department of Obstetrics and Pathology of Pregnancy, Medical University in Lublin, 20-081 Lublin, Poland

**Keywords:** cancer stem cells, endometrial cancer, CD133, CD117, CD44

## Abstract

**Simple Summary:**

In this manuscript, the characteristics of cancer stem cells and their potential role in endometrial cancer are explored. An understanding of the mechanisms behind cancer stem cells in endometrial cancer is crucial because these cells potentially play a key role in how the cancer grows and spreads. The authors aim to review the most recent data concerning cancer stem cells in endometrial cancer, especially biomarkers which could enable identification and prognosis. The determination of signalling pathways crucial for the functioning of cancer stem cells could help with the design of effective treatment strategies.

**Abstract:**

Endometrial cancer is one of most common types of gynaecological tumours in developing countries. It has been suggested that cancer stem cells play an important role in the development of endometrial cancer. These are a subset of highly tumorigenic cells with similar features to normal stem cells (unlimited proliferation, multi-potential differentiation, self-renewal, aggressiveness, invasion, recurrence, and chemo- and endocrine therapy resistance). Wnt/β-catenin, Hedghog, and Notch1 are the most frequently activated pathways in endometrial cancer stem cells. The presence of cancer stem cells is associated with the resistance to chemotherapy caused by different mechanisms. Various markers, including CD24, CD40, CD44, CD9, CD133, and CD 166, have been identified on the surface of these cells. A higher expression of such markers translates into enhanced tumorigenicity. However, there is no strong evidence showing that any of these identified markers can be used as the universal marker for endometrial cancer stem cells. Growing data from genomic and proteomic profiling shed some light on the understanding of the molecular basis of cancers in humans and the role of cancer stem cells. However, there is much left to discover. Therefore, more studies are needed to fully uncover their functional mechanisms in order to prevent the development and recurrence of cancer, as well as to enhance treatment effectiveness.

## 1. Introduction

Endometrial cancer is the most common type of gynaecological hormone-dependent tumour in developing countries [1]. According to American Cancer Society estimates, approximately 67,880 new cases of uterine (including endometrial) cancers will be diagnosed in 2024, and 13,250 women will die from this disease in United States [2]. In turn, according to GLOBOSCAN, the age-standardised incidence of endometrial cancer and patient mortality was 8.7 and 1.8 in 2020 [3].

The risk of endometrial cancer is increased among post-menopausal women [4,5]. The median age at diagnosis is 63 years [6,7]. Due to its characteristic symptoms, including post-menopausal uterine bleeding, endometrial cancer is usually diagnosed at an early stage [8]. Clinical data indicate that only one-third of patients have localised cancer at diagnosis [6]. The prognosis of early-stage (I and II) or localised endometrial cancer is favourable, and treatment can be limited to surgery, followed, in some cases, by brachytherapy and/or external beam radiation therapy [9]. High-risk patients with an early stage of endometrial cancer also undergo platinum-based chemotherapy [4]. However, endometrial cancer is frequently recurring and spreads to the vagina, pelvic and para-aortic lymph nodes, peritoneum, and lungs (distant metastases) [10,11]. Recurrences are reported in about 13% of high-risk patients, but also in 3% of low-risk patients [12,13]. The estimated five-year overall survival of patients with stage I disease is 95%, but drops to 69% in the case of patients with stage II disease [6]. The five-year survival rate of patients with advanced disease stages (III–IV) is as low as 15–17% [6,14]. The low survival in the advanced stages is due to the poor response of endometrial cancer to chemotherapy and radiotherapy. The standard first-line treatment involves surgery, followed by a combination of carboplatin and paclitaxel [15,16]. However, the quick development of treatment resistance and low response rates to single-agent chemotherapy, as well as to endocrine therapy with medroxyprogesterone acetate (MPA), limit the range of therapeutic options [4]. New drugs, including pembrolizumab alone (in microsatellite instability (MSI)-high tumours) or in combination with the angiogenesis inhibitor lenvatinib (in patients without MSI), have been approved by the US Food and Drug Administration (FDA) [17,18,19]. However, treatments that would considerably increase patient survival are awaited.

Cancer stem cells (CSCs) have been suggested to play an important role in the development of endometrial cancer [20]. Indeed, Hubbard et al. [20] identified a sparse population of endometrial cancer cells showing the ability differentiate and initiate tumours. CSCs were also found to express several genes with pluripotent features [21]. It is now believed that CSCs are at least partly responsible for drug resistance, tumour progression, invasiveness, and metastasis [10,22,23]. Therefore, novel approaches to treating tumours are based on inhibitors of the CSC signalling pathways, as well as adoptive cell therapy targeting specific CSC antigens and modulators of their epigenetic mechanisms [15,24,25,26,27].

## 2. Endometrial Cancer

### 2.1. Classification

First, the historical endometrial cancer classification was based on clinical and hormonal features. According to Bokhman’s dualistic model, sporadic endometrial cancer was categorised into oestrogen-dependent highly-to-moderately differentiated carcinomas with an endometrioid morphology (Type 1) and oestrogen-independent non-endometrioid carcinomas (Type 2) [28]. Type 1 carcinomas occur more frequently and are less aggressive (80% of patients in this group have grade 1 or 2 tumours) [29,30]. Moreover, type 1 cancers show a high expression of oestrogen and progesterone receptors, and a low potential for lymphovascular invasion, and are associated with a favourable prognosis [28]. By contrast, up to 66% of women with type 2 endometrial cancer have a high-grade (grade 3) tumour with a high potential for lymphovascular invasion, little progesterone sensitivity, and a poor prognosis. The revised staging system for endometrial cancer which was published in 2023 incorporates diverse histological classifications, tumour patterns, and molecular categorisation to more accurately depict the intricate characteristics of different endometrial carcinoma types and their underlying biological dynamics [31]. According to this new system, stage I includes various subcategories from IA to IVC based on the extent and nature of the endometrial carcinoma. In turn, the Cancer Genome Atlas (TCGA) classification divides endometrial cancer into four types based on the overall mutational burden (including phosphatase and TENsin homolog (PTEN), p53, polymerase epsilon (POLE) mutations, MSI, and histology), which translates into different clinical outcomes [15,32]. A recently developed prognostic classification system for endometrial cancer that uses next-generation sequencing of the most discriminant genes shows a nearly 100% accuracy [33]. Endometrial cancers carrying POLE proofreading mutations show a favourable prognosis despite a strong association with high-grade cancers [34]. Moreover, most endometrioid tumours have been demonstrated to carry few somatic copy number alterations [32]. However, cancers burdened with extensive somatic copy number alterations show considerably worse progression-free survival compared with other groups.

### 2.2. Risk Factors

Various mutations in different genes have been linked with the development and progression of cancers [35]. Mutations in genes including TP53 (71%), PIK3CA (31%), PPP2R1A (25%), ABCC9, CYP4X1, CHD4, MAP3K4, FBXW7, and SPOP have been identified as key factors in the pathogenesis of endometrial cancer [35,36]. Moreover, CHD4 has been demonstrated to be specific to endometrial cancer [35]. This epigenetic factor contributes to metastasis via the modulation of the EZH2/β-catenin axis and is associated with a poor prognosis [37]. The identification of cancer-promoting alterations could contribute to the development of novel diagnostic and treatment methods. The activation of specific signalling pathways translates into tumour behaviour and patients’ outcomes. For example, the upregulation of genes linked to the epithelial-to-mesenchymal transition (EMT), such as SNAI1 and TWIST1, has been shown to enhance the tumorigenicity of endometrial CSCs [38]. In turn, the higher paclitaxel resistance in endometrial cancer cells was linked to PI3K/AKT activation mediated by human epidermal growth factor receptor 2 (HER2) [39].

Apart from genetic alterations, polycystic ovary syndrome, diabetes, obesity, early menarche, late menopause, Lynch syndrome (in women below the age of 50 years), and infertility are also well-known risk factors for endometrial cancer [40,41,42]. Additionally, exposure to air pollutants, including polycyclic aromatic hydrocarbons (PAHs), is a risk factor for endometrial cancer [43]. The dose or concentration and duration of exposure to environmental pollutants have been suggested to affect the initiation of chemical carcinogenesis. PAHs not only activate environmental carcinogens but also stimulate cancer cell proliferation [44,45]. Indeed, growing evidence supports a relationship between exposure to PAHs and the development of CSCs that is mediated by the aryl hydrocarbon receptor (AHR) pathway and the activation of the WNT/β-catenin pathway and the ABCG2 transporter [43,46,47,48,49]. Increased levels of AHR mRNA were observed in early-stage endometrial cancer due to an oestrogen-related mechanism compared to advanced disease stages and normal endometrium. Moreover, the expression of AHR was found to be higher in oestrogen-independent responsive grade 3 endometrioid adenocarcinomas than in oestrogen-dependent responsive grade 1 and 3 endometrioid adenocarcinomas [50].

The development of tumours is also promoted by hypoxia, since hypoxic tension stimulates the initiation and maintenance of CSC stemness and contributes to the formation of a microenvironment that favours the survival and thriving of tumour cells [51]. Hypoxia-inducible factor (HIF)-1α triggers a series of adaptive responses through decreasing oxygen consumption and the release of relevant stimulators including glycolytic enzymes, vascular endothelial growth factor (VEGF), and erythropoietin. It also induces the expression of CD44 and CD133, MYC, homeobox protein (NANOG), octamer binding transcription factor 4 (OCT4), and SRY-box transcription factor 2 (SOX2), and promotes EMT [51]. The interaction between HIF-1 and the Notch pathway has been found to enable the maintenance of tumour cells under hypoxia [52,53].

## 3. Stem Cells and Endometrium

The human endometrium contains populations of epithelial and stromal colony-forming cells and has an exceptional regenerative capacity [54,55]. The endometrium undergoes a monthly cycle of growth, proliferation, differentiation, shedding, and regeneration under the influence of the circulating ovarian steroid hormones oestrogen and progesterone [56,57,58]. Endometrium regeneration with each menstrual cycle as well as the rapid enlargement and adjustment of the uterus to accommodate the developing foetus are possible, owing to the presence of stem cells.

Stem cells represent unspecialised cells within the human body, possessing the capacity for differentiation into diverse cellular lineages observed throughout the continuum of life in both embryonic and adult tissues [59]. They play essential roles in neonatal development and tissue regeneration following injury or disease by serving as the primary source for specific cell types within tissues and organs due to the ability to migrate freely throughout different tissues [60,61]. Stem cells undergo a series of specialisation stages during which their developmental potential diminishes progressively. Totipotent stem cells, with the highest differentiation potential, can generate cells for the entire organism. Pluripotent stem cells, such as embryonic stem cells (ESCs), form germ layer cells but not extraembryonic structures. Multipotent stem cells, including haematopoietic stem cells, specialise in specific cell lineages but possess a broader differentiation range than oligopotent stem cells, which can differentiate into several cell types. Unipotent stem cells, with their restricted differentiation capabilities, possess the unique ability to repeatedly divide, rendering them promising candidates for regenerative medicine applications [59]. Endometrial stem cells are distinguished by their extraordinary capacity for self-renewal and differentiation into assorted cell phenotypes, including stromal, epithelial, and vascular lineages [62]. Endometrial tissue harbours diverse stem cell populations, including epithelial-like, stromal-like, and perivascular endometrial stem cells, each characterised by distinct molecular and functional attributes [62].

The fundamental traits of stem cells involve their capacity for self-renewal, proliferation, and high differentiation capacity (pluripotency), ensuring a continual generation of progeny aimed at replacing aging, senescent, or impaired cells [61,63]. These characteristics distinguish them from other types of cells. Stem cells have been demonstrated to be involved in the telomere maintenance mechanism. Telomeres, comprised of repetitive DNA sequences, undergo gradual attrition with each successive cellular division as a result of DNA polymerases’ inability to completely replicate a linear template [64]. Upon reaching a critical threshold of shortening, telomeres activate a DNA damage response pathway, culminating in cell cycle arrest [65]. However, pluripotent stem cells possess mechanisms for telomere elongation, thereby protecting their prolonged proliferation and self-renewal capabilities [66]. However, excessive telomere elongation can prove detrimental, prompting the activation of a rapid telomere deletion process known as telomere trimming [67]. This delicate balance between telomere elongation and trimming mechanisms is essential for maintaining genomic stability and cellular homeostasis in pluripotent stem cells [68].

Since stem cells exhibit an extended lifespan compared to somatic cells, the likelihood of accumulating genetic alterations, potentially predisposing them to malignant transformation, is increased. The accrual of abnormalities and mutations within endometrial stem cells may trigger the initiation and progression of various endometrial pathologies, including endometrial cancers [69]. The occurrence of a few mutations can disrupt the regulatory mechanisms governing self-renewal and proliferation, fostering the emergence of cancerous phenotypes. Epithelial-like endometrial stem cells have been suggested to contribute to the formation and progression of endometrial malignancies not only as a result of genetic instability but also as a result of the enhanced proliferative capacity [70]. The accumulation of genetic mutations within epithelial-like stem cells was suggested to trigger the transformation of normal endometrial stem cells into malignant ones—cancer stem cells [71].

## 4. Cancer Stem Cells

CSCs are a subset of highly tumorigenic cells that show similar features to normal stem cells including unlimited proliferation, self-renewal, aggressiveness, multi-potential differentiation, recurrence, invasion, metastasis, chemoresistance, and endocrine therapy resistance [10]. Self-renewal means that CSCs are able to regenerate and give rise to a subset of cells displaying an abnormal differentiation and survival potential despite apoptotic signals, which is crucial for tumour maintenance [72]. The predisposition of endometrial CSCs to spread to other organs is associated with their ability to migrate, stimulate angiogenesis, and release the extracellular matrix [72]. CSCs were identified for the first time in acute myeloid leukaemia but were later also observed in other types of tumours [73]. Subsequent studies demonstrated that CSCs form spherical colonies expressing surface markers and executing biological functions [74]. Endometrial CSCs with the ability to form sphere-like structures were found to have a greater self-renewal potential and capacity for chemoresistance. Moreover, their presence favours tumour initiation [75]. The results of one study demonstrated that endometrial CSCs account for 0.02–0.08% of cells in endometrial cancer cell lines and 3.4% in primary tumours [20].

CSCs contribute to tumour initiation and development since they constantly give rise to numerous descendant cancer cells [16]. They are involved in every step of tumour formation, development, and metastasis. These cells are capable of switching between higher oxidative mitochondrial metabolism and anaerobic glycolysis [76]. They are also capable of undergoing phenotypic transition following stimulation (plasticity) [77]. The results of some studies have revealed that CSCs are characterised by increased motility, as well as enhanced migratory and invasive properties, compared with normal stem cells [78,79]. These cells show the ability to detach from the original tumour, migrate, and invade a new site, which explains their high metastatic potential [16]. The presence of high amounts of CSCs is associated with enhanced aggressiveness and poor outcomes. CSCs use ATP-dependent efflux pumps, such as multidrug resistance protein 1 (MDR1), which enable their survival in unfavourable conditions and make the cancer unresponsive to chemotherapy [80]. In addition to the activity of ATP binding cassette (ABC) efflux transporters, other resistance mechanisms in endometrial CSCs include resistance to DNA damage, aldehyde dehydrogenase (ALDH) activity, the activation of developmental pathways and microenvironmental stimuli, resistance to apoptosis, and autophagy [22,55,72].

Several hypotheses have been formulated concerning the formation of CSCs. According to one hypothesis, CSCs arise from normal/adult stem cells via the ongoing acquisition of epigenetic alterations and genetic mutations [81]. Another thesis concerning CSC formation (the “big bang” model) states that, following the initial transformation, cells with initiating mutations grow predominantly to form a single expansion populated by numerous intermixed sub-clones [82]. The growth of tumours is an evolutionary process; thus, it seems that information on their early development is encoded in the genome and is responsible for intra-tumour heterogeneity. New sub-clonal mutations are constantly generated as a result of replication errors; however, only the earliest are omnipresent, while later alterations appear increasingly in only smaller tumour sub-populations. Some sub-clonal mutations acquired during growth may confer survival advantages. Therefore, the timing of a mutation, rather than clonal selection, was suggested to determine tumour pervasiveness [82].

Mani et al. [83] suggested that CSCs are derived from differentiated cancer cells via EMT-mediated dedifferentiation. EMT regulates the expression of genes involved in various metabolic pathways and metabolic reprogramming [84]. The activation of EMT was demonstrated to be crucial for the formation of CSCs and is involved in the acquisition of a mesenchymal phenotype and loss of epithelial features. This process enables malignant cells to metastasise from a primary tumour [85]. The nuclear expression of EMT transcription factors, such as SLUG and TWIST in endometrial CSCs, links EMT with stemness and the expression of programmed death ligand 1 (PDL1), which enables immune evasion [86]. Numerous studies have shown that PDL1 expression affects stem-like characteristics, including cell growth, stemness, metastasis, and drug resistance [87,88,89]. The proto-oncogene MYC directly promotes PDL1 expression, thus contributing to immunosuppression. In turn, MYC downregulation results in immune cell infiltration [90]. In general, the MYC family regulates various cellular processes and is involved in tumorigenesis [91]. The co-expression of MYC and CD44 has been observed in endometrial CSCs [92]. The expression of MYC appeared necessary for SALL4-induced EMT, invasion, and resistance to antineoplastic drugs in endometrial cancer cells [93].

The formation of CSCs involves the expression and release of “stemness” molecules [94]. Guy et al. [75] observed the enhanced expression of genes related to stemness, including OCT4A, MYC, KRT18 (which encodes CK-18), SOX2, B lymphoma Mo-MLV insertion region 1 homolog (BMI1), ABCG2, NANOG, and NES (which encodes nestin) in CD133-positive and CXCR4-positive endometrial cancer cells. The results of other studies have demonstrated that the expression of OCT4 and SOX2 in endometrial CSCs could be associated with their self-renewal capacity [20,95]. Endometrial CSCs isolated from endometrial cancer expressing stemness markers, including SOX2, OCT4, MYC, ABCG2, NES, and NANOG, have been demonstrated to show greater expansion potential and colony-forming capabilities, as well as to express self-renewal genes [20,96]. OCT4, SOX2, and NANOG were suggested to contribute to pluripotency [97].

In endometrial cancer cells, stemness is modulated mostly via the WNT, Notch (which controls cell development and stimulation-triggered differentiation), and Hedgehog pathways [98]. Numerous studies have confirmed the activation of CSC-related signalling pathways, including WNT/β-catenin and Notch in various tumours, including endometrial cancer [99,100,101,102]. The self-renewal and maintenance of CSCs is also associated with the activation of the WNT/β-catenin, Notch, BMI1, and SHH pathways [103,104,105,106]. The WNT/β-catenin pathway controls endometrial proliferation and differentiation [107]. The downregulation of WNT signalling in endometrial cancer cells hindered CSC proliferation and migration, confirming the importance of this pathway in tumour development [96]. As aforementioned, Notch signalling has been found to be involved in CSC proliferation and viability. The use of various micro-RNAs (miRNAs) targeting the Notch pathway hindered the proliferation of the CD44/CD133-positive subpopulation in endometrial cancer (miRNA134), limited the expression of vimentin (VIM), reduced EMT, and decreased tumour formation and invasiveness (miRNA-34a) [108,109]. In turn, the Hedgehog signalling pathway is involved in cell differentiation and growth via the transmembrane protein smoothened (SMO) and the glioma transcriptional factor GLI1. Sonic hedgehog (SHH) and SMO were found to be overexpressed in endometrial cancer compared with endometrial hyperplasia and normal endometrial epithelium [110].

The PTEN-PI3K/AKT/mammalian target of rapamycin (mTOR) pathway is another important element contributing to CSC stemness via the upregulation of EMT triggers, including enhancer of zeste homolog 2 (EZH2), BMI1, SNAI1, and SLUG [15,111]. Other pathways involved in the upregulation of self-renewal and differentiation in CSCs include AKT, mTOR, Janus kinase (JAK)/signal transducer and activator of transcription (STAT), and nuclear factor-κB (NF-κB) [112,113,114].

Numerous studies have reported a correlation between the presence of CSCs and resistance to chemotherapy [43,115]. The upregulation of the drug efflux mechanism that enables drug efflux against the concentration gradient and results from a high expression of several specific transporter proteins, such as ABC drug transporters, especially ABCG2, is partly responsible for this phenomenon [116]. Liu et al. [93] demonstrated that the downregulation of MYC in endometrial cancer decreased cell invasion and drug resistance considerably.

The problems with cancer treatment have resulted in numerous studies trying to identify plausible new therapeutic targets. In one of them, Cao et al. [10], based on a tandem mass tag quantitative proteomic analysis, identified 5735 proteins in spheroid cells from endometrial cancer cell lines. They are observed to have an elevated expression of hexokinase 2 (HK2) and 6-phosphofructo-2-kinase/fructose-2,6-biphosphatase 3 (PFKFB3) (both of which encode key molecules in the HIF-1 pathway), as well as G Protein-coupled receptor class C group 5 member A (GPRC5A) in spheroid cells. The authors demonstrated that knockdown of HK2 in endometrial cancer cells reduced cell proliferation and the self-renewal ability [10]. The results of other studies indicated a role for PFKFB3 in the regulation of endothelial glycolysis, the promotion of angiogenesis, and the metastasis of malignant tumours [117,118]. PFKFB3 inhibition limited tumour invasion, intravasation, and metastasis [119]. Moreover, this was found to be unique for CSCs and enabled their differentiation from both non-stem cells and induced pluripotent stem cells, specifically under hypoxia [120]. In turn, GPRC5A was demonstrated to stimulate the self-renewal and metastasis of bladder CSCs [121,122].

Potential therapeutic targets are presented on Figure 1.

As aforementioned, the treatment of endometrial cancer is challenging. Figure 2 presents novel therapies that are being assessed for safety and efficacy in clinical trials.

## 5. CSC Markers

Cluster of differentiation (CD) markers, such as CD24, CD40, CD44, CD9, CD133, and CD 166, have been identified on the surface of CSCs [23,43,123]. An analysis of spheroid cells from endometrial cancer cell lines (Ishikawa and HEC1A) demonstrated the enhanced expression of CD90, CD117, CD133, and W5C5. In turn, other markers, such as CD29, CD44, and CD105, were found to not be uniquely expressed in these cells [10]. In endometrial cancer, specific cell surface markers, including CD44-antigen (CD44) and prominin-1 (CD133), have been identified as surrogate markers for CSCs [58]. These surface markers are frequently used to identify endometrial-carcinoma-derived stem-like cells [8]. The expression of these markers has been linked to tumorigenicity, invasiveness, and metastasis [124,125]. Moreover, CSCs exhibit high ALDH activity [126,127]. The increased expression of CSC-related markers, including CD44, CD133, ALDH1A1, and NANOG, in spheroid cells of endometrial cancer translates into enhanced tumorigenicity [55,128].

### 5.1. CD133

CD133 is a pentaspan transmembrane glycoprotein, the sub-cellular localisation of which enables its interaction with lipid rafts participating in the signalling cascade [129]. Mizrak et al. [130] suggested that this molecule could be involved in cell membrane organisation. Jaksch et al. [131] showed that the expression of CD133 in both CSCs and normal cells depended on the cell cycle. Various studies indicated that CSCs expressing CD133 comprised 5.7–27.4% of cells in primary endometrial tumours and their presence was associated with a worse prognosis [75]. The high expression of CD133 on the surface of CSCs correlates with a greater self-renewal ability, higher proliferative potential, lymphovascular invasion, and a poor prognosis [23,129]. Rutella et al. [75] demonstrated in a mouse model that cells expressing CD133 showed an increased ability to aggressively proliferate and form colonies compared with CD133-negative cells [75]. Moreover, CD133-positive cells had a greater ability to migrate from the primary mass to blood vessels and other organs and a higher resistance to chemotherapy than CD133-negative cells. Rutella et al. [75] suggested that CD133-positive populations of endometrial cancer cell lines form floating spheres and colonies that originate from clonal proliferation. Additionally, Kyo et al. [132] provided evidence that CD133-positive cells have a higher colony-forming ability in vitro and enhanced tumorigenicity in immunocompromised mice. A high CD133 expression in specimens of endometrioid-type endometrial cancer was associated with decreased overall survival compared with a low CD133 expression [132]. In addition to the CD133 expression, the FIGO stage and histological grade negatively influenced overall survival rates. The CD133 expression (hazard ratio (HR) 3.90; *p* < 0.045) and FIGO stage (HR 3.94; *p* < 0.042) were independent predictive factors for patient survival [132]. The expression of CD133 translates into higher rates of endometrial tumour relapse and worse survival [54,75,133]. A high expression of TGFB1 has been observed in the CD133-positive CSC subpopulation of endometrial cancer cells. TGF-β1 was found to activate EMT to trigger metastasis [75]. The TGF-β1/CD133 pathway was found to stimulate cell invasion, stem-like characteristics, and therapy resistance [134]. Moreover, TGF-β1 can stimulate genomic instability via hindering the repair of DNA double-strand breaks [135,136]. Nakamura et al. [137] suggested that CD133 was a risk factor for endometrial cancer due to the higher proliferative and tumorigenic potential and cisplatin and paclitaxel resistance. The results of another study revealed that CD33-positive cells have increased tumorigenic potential [125]. Aggressive tumour behaviour has also been linked with the upregulation of ABCG2 and matrix metalloproteinases (MMPs) in CD133-positive cells [132]. The expression of MMP14 (which encodes MT1-MMP) increases the invasive capacity of CD133-positive cells, whereas the knockdown of MMP14 with small interfering RNA (siRNA) mostly abolishes this effect.

The expression of CD133 in endometrial CSCs was found to be frequently accompanied by the upregulation of CD44 and NES, which enhanced proliferation and infiltration [75,124,138]. Bokhari et al. [139] confirmed that NES knockdown hindered cell growth and limited the invasive potential and colony formation abilities of endometrial cancer cell lines, whereas its overexpression was associated with a malignant phenotype.

### 5.2. CD44

CD44 is a transmembrane adhesion molecule suggested to be a useful marker of CSCs in endometrial cancer. This molecule is also involved in cancer invasion and metastasis [125,133]. CD44 expression was found to positively correlate with PDL1 expression in various tumours, as well as with immune infiltration [140]. This adhesion molecule has been demonstrated to be involved in tumour cell invasion and metastasis in endometrial cancer cell lines [133]. Zagorianakou et al. [141] linked CD44 expression with an enhanced proliferation of endometrial carcinomas of the endometrioid type, a higher tumour grade, and progesterone receptor status. Park et al. [124] found a close relationship between CD133 or CD44 expression and endometrial cancer progression and a poor prognosis.

### 5.3. SMOC2

Lu et al. [58] suggested that SPARC-related modular calcium binding 2 (SMOC2) could also be used as a signature gene of endometrial CSCs. SMOC2 protein, the levels of which are enhanced during embryogenesis and wound healing, shows angiogenic activity and is capable of stimulating endothelial cell proliferation and migration [58,142,143]. Lu et al. [58] found that SMOC2 expression was increased in sphere cultures and in CD133/CD44-positive cells compared with CD133/CD44-negative cells and also boosted the chemoresistance of endometrial cancer cells. Furthermore, SMOC2 silencing resulted in the diminished clonogenic potential of endometrial cancer and reduced the expression of stemness-related genes, including SOX2, OCT4, and NANOG [58]. In endometrial cancer, the SMOC2-related activation of WNT/β-catenin signalling promoted the progression of endometrial cancer since this pathway is vital for cell growth, differentiation, proliferation, and survival [58,144,145]. Furthermore, this pathway is involved in the formation and maintenance of stem cells and CSCs [144].

### 5.4. CXCR4

The invasive and metastatic phenotypes of CSCs have also been suggested to be mediated by the chemokine ligand/receptor axis [132]. In endometrial cancer, CXC motif chemokine receptor 4 (CXCR4) was demonstrated to be highly expressed by some tumour cells [146]. The activation of CXCR4, a stromal cell-derived factor-1 receptor, triggers signalling pathways promoting increased survival, enhanced proliferation, the degradation of the extracellular matrix, drug resistance, and angiogenesis in malignant tumour cells [98]. The expression of both CXCR4 and CD133 has been reported in all types of human tumours, including endometrial cancer [138]. Moreover, CXCR4 was found to be upregulated considerably in endometrial tumours compared with atypical, simple hyperplasia and normal endometrial tissue [125]. The results of several studies have found that the CXCL12/CXCR4 axis is partly responsible for tumour progression, angiogenesis, metastasis, and survival [147]. The actions of CXCR4 are associated with the induction of signalling pathways involved in gene transcription, survival, proliferation, and chemotaxis. Sun et al. [138] demonstrated that CD133/CXCR4-positive endometrial cancer cells comprised less than 10% of the total population, which is in agreement with the findings that CSCs constitute only a small percentage of cells in malignant tumours. Cioffi et al. [148] demonstrated considerably lower two-year survival rates in patients with a high expression of CD133 and CXCR4 compared with patients with a low CD133 and CXCR4 expression. The growth of CD133/CXCR4-positive cells in vitro was greater than that of CD133/CXCR4-negative cells, indicating enhanced proliferative properties. Sun et al. [138] also revealed that CD133/CXCR4-positive cells have an increased potential to form more spheres and colonies, which appear to be typical characteristics of stemness in various tumours [149,150]. CD133/CXCR4-positive cells also exhibited a higher expression of stemness genes [138]. The injection of CD133CXCR4-positive cells into nude mice was associated with tumour formation, and tumour formation was not observed following inoculation with CD133/CXCR4-negative cells [138].

### 5.5. CD117

CD117 (c-kit) is a cell surface receptor tyrosine kinase that has been proposed as a CSC marker in various tumour types [15,43]. Following stimulation by stem cell factor, CD117 triggers cell replication, differentiation, and survival, and CSCs acquire stemness properties [151,152,153]. The results of some studies have demonstrated that endometrial CSCs expressing CD117 have increased the proliferative and colony-forming potential [15,154]. Zhang et al. [154] suggested that a high CD117 expression could be used as an independent prognostic factor. In addition to endometrial cancer, the expression of CD117 in CSCs has also been demonstrated in ovarian and lung cancer [152,155].

### 5.6. CD55

CD55 is an intrinsic cell surface complement inhibitor [43]. A high expression of CD55 has been reported in endometrial cancer cells and CSCs. Cells expressing this inhibitor displayed a higher self-renewal ability, as well as a higher resistance to chemotherapy, than CD55-negative cells [156]. CD55 expression in endometrioid ovarian and endometrial CSCs was demonstrated to enhance self-renewal properties and cisplatin resistance. In endometrial endometrioid cell lines, the inhibition of CD55 with saracatinib was associated with cisplatin resensitisation [156].

### 5.7. ALDH1

ALDH1 was demonstrated to be highly active during the early steps of stem cell differentiation [157,158]. Endometrial cancer cells with a high ALDH1 expression were found to be more tumorigenic and invasive, as well as resistant to cisplatin therapy and associated with a worse prognosis in patients with endometrial cancer (*p* = 0.01 for overall survival), compared with cells with a low expression of ALDH1. Additionally, Rahadiani et al. [159] observed that cells with a higher ALDH1 expression had higher tumorigenic potential, as well as increased invasiveness and resistance to cisplatin, compared with cells with a low ALDH1 expression. These features were associated with a poor prognosis in patients with endometrial cancer [159]. The expression of ALDH and CD133 in debulked primary tumour specimens was found to correlate with poor patient survival [157]. Mori at al. [160] observed that ALDH-dependent glycolytic activation mediated stemness and chemoresistance in spheroid uterine endometrial cancer derived from patients. The disulfiram and N,N-diethylaminobenzaldehyde (DEAB)-related inhibition of ALDH1 decreased the proliferation of spheroid CSCs [128]. Ran et al. [161] suggested that the ALDH expression in endometrial CSCs mediates autophagy, which contributes to CSC survival and chemoresistance, since they observed a decrease in cell growth and self-renewal potential following the use of an autophagy inhibitor (3-methyladenine or chloroquine).

### 5.8. NANOG

NANOG expression was suggested to be useful as a diagnostic marker, enabling the differentiation between true dysplasia and reactive lesions [23,162]. An increased expression of NANOG has been detected in precancerous lesions (high-grade dysplasia), as well as in cancerous tissue. In endometrial CSCs, NANOG expression was found to be regulated by transcription factor 3 (TCF3), OCT4, and SOX2 [15]. Grubelnik et al. [163] suggested that the high expression of NANOG is associated with a more advanced cancer, a higher cancer grade, resistance to treatment, and, thus, a worse prognosis. NANOG has also been suggested to be involved in the acquisition of stem-cell-like features, including self-renewal and immortality [23]. Al-Kaabi et al. [23] demonstrated the expression of NANOG in 88.37% of endometrial carcinoma cases. Moreover, they reported that NANOG expression correlated with deep myometrial invasion, a higher cancer grade, and a positive lymph node status, all of which indicated a worse prognosis. According to other studies, a high NANOG expression translates into the presence of poorly differentiated, advanced tumours and poor patient survival [164,165,166].

### 5.9. Other

Endometrial CSCs have also been reported to express the neuroendocrine marker synaptophysin [8]. Helweg et al. [8] observed that nuclear synaptophysin, CD133, CD44, and Nestin indicated endometrial-CSC-like characteristics. Additionally, the levels of CK-18, a structural protein normally present in many single-layer epithelia, were demonstrated to correlate with the clinical stage, number of positive lymph nodes, metastasis, and recurrence, as well as poorer overall and disease-free survival [167].

The utility of the aforementioned markers has been suggested in many studies; however, Tabuchi et al. [168] demonstrated that CSC populations in uterine endometrioid adenocarcinoma showed functional heterogeneity and suggested that there was no distinctive biomarker for the identification of CSCs as various colonies within the same tumour could express different markers. Spherical clones were found to be less tumorigenic but more resistant to chemotherapy. In turn, leukaemia-like clones show opposing features. Therefore, more studies are required to understand the functional mechanisms of CSCs and to target these cells in order to improve the effectiveness of cancer treatment and prevent its recurrence.

Table 1 summarises studies focused on possible interventions related to CSCs in endometrial cancer.

## 6. Conclusions and Future Directions

Cancer stem cells have recently emerged as a crucial target for cancer treatment. However, the available knowledge has been complicated by recent insights revealing the significant influence of the tumour microenvironment in modulating the properties of cancer stem cells (CSCs). This is of particular importance as the transformation of non-tumorigenic cancer cells into CSCs has been observed, leading to potential variations in the CSC population within a neoplasm over time. The plausible heterogeneity of CSCs justifies the existence of multiple biomarkers. Conflicting results from various studies regarding the diagnostic efficacy of these biomarkers highlight the necessity for thorough validation in order to develop successful, biology-oriented therapeutic strategies. Currently, only a limited number of clinical trials targeting CSCs with drugs have been initiated. While early-phase studies indicate the safety of these drugs, there is a lack of comprehensive data on their therapeutic efficacy.

## Figures and Tables

**Figure 1 cancers-16-01083-f001:**
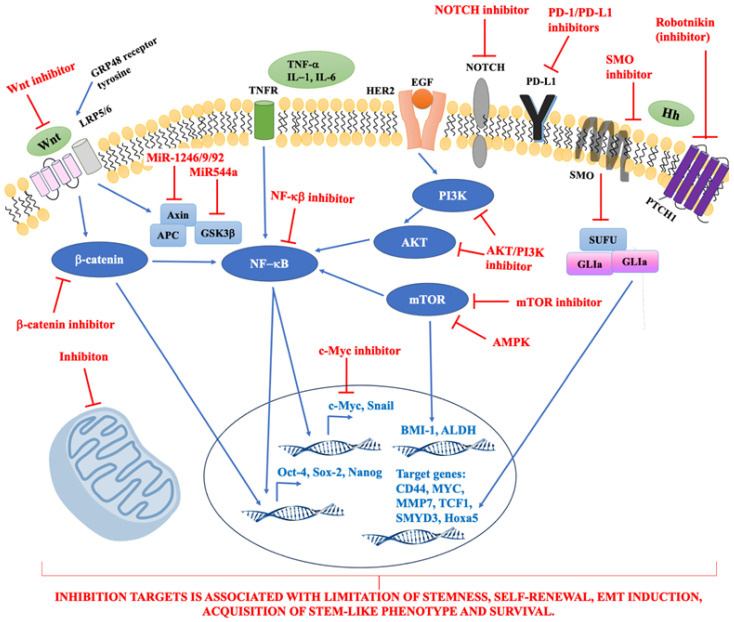
The summary of potential therapeutic targets. Abbreviations: ALDH, aldehyde dehydrogenase; AMPK, 5′AMP-activated protein kinase; BMI-1, B-lymphoma Mo-MLV insertion region 1; CD44, cluster of differentiation 44; EGF, epidermal growth factor; GLI, transcription factor GLI; GRP48, G Protein Coupled Receptor 48; GSK3β, glycogen synthase kinase-3β; HER2, human epidermal growth factor receptor 2; Hh, hedgehog signalling pathway; Hoxa5, homeobox A5; IL-1,-6, interleukin-1,-6; LRP5/6, Low-density lipoprotein receptor-related protein 5; MMP-7, Matrix Metalloprotease-7; mTOR, mammalian target of rapamycin; MYC, MYC MYC transcription factor; NF-κB, nuclear factor κB; PD-1, programmed death receptor-1; PD-L1, programmed death receptor-1 ligand; PI3K, phosphatidylinositol 3-kinases; PTCH1, receptor called Patched 1; SMO, signal transducer called Smoothened; SMYD3, SET and MYND domain containing 3; SUFU, cytoplasmic protein; TCF1, T cell factor 1; TNF-α, tumour necrosis factor α; TNFR, tumour necrosis factor receptor; Wnt, Wnt signalling pathway.

**Figure 2 cancers-16-01083-f002:**
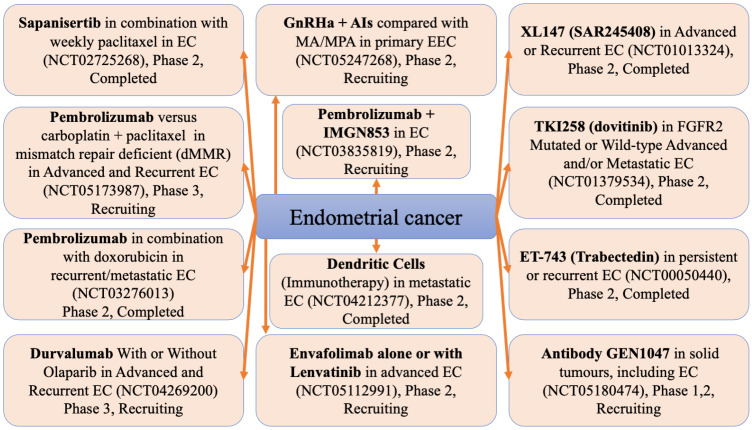
Clinical trials of new endometrial cancer therapies. AIs, aromatase inhibitors; EC, endometrial carcinoma; EEC, early endometrial carcinoma; GnRHa, gonadotropin-releasing hormone analogue; IMGN853, mirvetuximab soravtansine; MA, megestrol acetate; MPA, medroxyprogesterone acetate.

**Table 1 cancers-16-01083-t001:** The summary of possible therapeutic interventions targeting endometrial cancer CSC described in the literature.

Population	Type of Intervention	Results	Ref.
HEC1 cells and RK12V cells, which have features of CSCs	Salinomycin	- Reduction of fibronectin expression - Induction of apoptosis and inhibition of WNT signalling - Inhibition of proliferation, migration, invasiveness, and tumorigenicity of CSCs	[96]
(a) Ishikawa (ER/PR-positive and KLE (ER/PR-negative) cell lines (b) 12 patients with endometrial cancer	(a) 30.4 μM medroxyprogesterone 17-acetate (MPA) for 6 days (b) Progestin	- Significant reduction in percentage of live cells (Ishikawa *p* = 0.036; KLE *p* = 0.0002), increased apoptosis (Ishikawa *p* = 0.01; KLE *p* = 0.0006), and significant decrease in CD133-positive populations (Ishikawa *p* < 0.0001; KLE *p* = 0.0001) - Reduction in CD133 expression in patients who had a histological response to progestin treatment	[98]
Ishikawa and HEC1A cell lines	Metformin (0.5 mM or 1 mM)	- 1.7-fold reduction in the proportion of HEC1A CSCs (identified by high ALDH activity) by metformin (*p* ≤ 0.05 for 1 mM) - Significant reduction in the number of CD133-positive endometrial CSCs in a dose-dependent manner - Decreased CSC activity, but no effect on self-renewal capacity - Significant reduction in viability of Ishikawa cells (1 mM metformin) - Reduction in expression of genes associated with pluripotency and self-renewal in a dose-dependent manner	[169]
(a) HEC1B endometrial cancer stem cell line (b) BALB/c nude mice	(a) miR-34a mimics (b) agomiR-34a	- Significant decrease in cell viability, number of cell colonies, and number of migratory and invasive cells in the miR-34a mimics group compared to the miR-34a NC group - Overexpression of miR-34a reversed EMT-associated phenotypes - miR-34a acted as a tumour suppressor in vitro - miR-34a overexpression significantly suppressed tumour growth, decreased NOTCH1 expression, and reversed EMT-associated phenotypes - High expression of miR-34a showed anti-tumour efficacy in vivo	[109]
4-week-old BALB/c athymic nude mice	MONC-positive and MONC/miR-636-positive endometrial CSCs, Ishikawa cells, and HEC1A cells injected subcutaneously into nude mice	- MONC overexpression inhibited the expression of SNAI1, vimentin (VIM), and N-cadherin (CDH2) - MONC overexpression promoted expression of E-cadherin (CDH1) - Smaller tumour volume in MONC-positive and the MONC-positive and miR-636-positive groups vs. control - Smallest tumour volume in MONC-positive group - Inhibition of endometrial cancer tumour growth by MONC - miR-636 rescued MONC from endometrial cancer tumour growth inhibition	[170]
HHUA human endometrial cancer cell line and 4–5-week-old female ICR null/null (nude) mice	Four putative CSC-selective drugs, including imatinib, 10 µM; niclosamide, 0.5 µM; sorafenib, 10 µM; genistein, 40 µM; tranilast, 50 µM; itraconazole, 10 µM; celecoxib, 100 µM; lithium carbonate, 40 µM; retinyl acetate, 40 µM; temsirolimus, 20 µM; thalidomide, 1 µM; bevacizumab, 1 µM; ramucirumab, 1 µM; and everolimus, 0.2 µM	- Sorafenib and niclosamide significantly reduced the proportion of HHUA-SP cells - Sorafenib exhibited the greatest CSC-selective inhibitory effects on the proliferation, survival, invasion, and in vivo tumorigenesis of HHUA cells among the three categories of drugs evaluated - Sorafenib targeted signalling pathways associated with tumorigenesis, including RAF1-mediated cell proliferation and ZEB1-mediated EMT - Sorafenib was found to be an effective treatment for endometrial cancer with RAS mutations or upregulated RAF/ERK pathway activity	[16]
6-week-old female athymic nude (nu/nu) mice (SLAC, Shanghai, China), human endometrial cancer and tissues, and normal endometrium	Lenti-*SMOC2*-short hairpin RNA/AN3CA cells and Lenti-vector as a control at 5 × 10^6^ cells in 100 μL serum-free DMEM	- Silencing SMOC2 in endometrial cancer cells suppressed cell growth at high density in vitro and reduced xenograft tumour growth in vivo	[171]

Abbreviations: ALDH, aldehyde dehydrogenase; CSCs, cancer stem cells; DMEM, Dulbecco’s Modified Eagle Medium; EMT, epithelial-to-mesenchymal transition; ER, oestrogen receptor; HEC1, human endometrial adenocarcinoma cell line 1; HHUA-MP, main population of endometrial cancer cell line derived from a well-differentiated endometrial adenocarcinoma; HHUA-SP, side population of HHUA cell line; miR-34a NC, miR-34a negative control; MONC, lncRNA MONC, mir-99a-let-7c cluster host gene; PR, progesterone receptor; RK12V, rat endometrial cells expressing oncogenic human K-Ras protein.

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
