# Peer review of "Characteristics of Cancer Stem Cells and Their Potential Role in Endometrial Cancer"

_cancers, 2024, doi:10.3390/cancers16061083_

Round 1

Reviewer 1 Report

Comments and Suggestions for Authors

FrÄ…szczak and co-worker presented a review manuscript on "Cancer stem cells characteristics and their potential role in endometrial cancer". This review focussed on the most recent data concerning cancer stem cells in endometrial cancer, especially biomarkers that could enable identification and prognosis. This is very interesting and the manuscript needs minor revision before the acceptance.

1. The statistics of Endometrial cancer presented as per year 2019, that is quite old, hence, authors needs to present the statistical data as per the year 2023 report. 

2. One figure must be included to represent the relationship between the role of stem cell therapy in combating endometiral cancer through molecular mechanism.

3. Grammatical and typos must be rechecked before submission of this minor revision 

Author Response

We would like to thank for your valuable comments which helped to improve this manuscript. Your suggestion was taken into consideration and appropriate information was provided. New/corrected parts are marked to facilitate the assessment of changes. We did our best to fulfil your expectations and we hope that you will be satisfied with our corrections. All the corrections are made in the track changes mode.

FrÄ…szczak and co-worker presented a review manuscript on "Cancer stem cells characteristics and their potential role in endometrial cancer". This review focussed on the most recent data concerning cancer stem cells in endometrial cancer, especially biomarkers that could enable identification and prognosis. This is very interesting and the manuscript needs minor revision before the acceptance.

  1. The statistics of Endometrial cancer presented as per year 2019, that is quite old, hence, authors needs to present the statistical data as per the year 2023 report. - No report from 2023 is available for endometrial cancer, therefore, data concerning uterine cancer (95% of cases in this type of tumour is ascribed to endometrial cancer) were used
  2. One figure must be included to represent the relationship between the role of stem cell therapy in combating endometrial cancer through molecular mechanism. The figure summarizing potential therapeutic targets have been added
  3. Grammatical and typos must be rechecked before submission of this minor revision  We checked the text again to correct all grammar mistakes and typos

Reviewer 2 Report

Comments and Suggestions for Authors

This is a review, not a scientific article, therefore aimed at a wide audience of readers, where many of them want to know and understand. The theme of the review should be introduced to the reader in a step-by-step manner, following an almost historical chronology and with logical and clear explanations.

 The authors have created a work that is both compact and verbose, sometimes bordering on chaotic. The authors often report a collage of short sentences taken from many scientific works, which does not allow the reader a continuity of thought.

The authors should divide the review into sub-paragraphs. In terms of the sub-paragraphs, I highly recommend including a dedicated section that focuses on the fascinating topic of stem cells. Specifically, it would be beneficial to explore their unique characteristics as normal cells and delve into their remarkable capacity to undergo a transformation from normal cells to cancerous cells. In fact, the use of stem cells help to repair or replace damaged tissues and organs. They possess the ability to rescue us from diseases that can only be managed with drugs that treat the symptoms. On the other hand, they have also another function in our lives, one that is not as beneficial. They may indeed be the source of some, and perhaps most, cancers.

Stem cells possess the remarkable ability to renew themselves, migrate freely throughout different tissues, and transform into a wide range of specialized cells. However, stem cells can undergo malignant transformation when these processes are disrupted by gene mutations or certain circumstances. Besides their current focus, the authors should also allocate some attention towards discussing the role of stem cells in telomere maintenance mechanism.

The expansion of CSC (cancer stem cells) is commonly linked to the process known as epithelial-mesenchymal transition (EMT), which is considered a general mechanism for the metastasis of cancer cells and even certain virus-infected cells, not only for endometrial cancer. It is important to note that these suggestions are merely recommendations, however, it would also be beneficial for the authors to emphasize also the various functions of stem cells, as well as in relation to the considerations surrounding endometrial cancer.

Author Response

We would like to thank for your valuable comments which helped to improve this manuscript. Your suggestion was taken into consideration and appropriate information was provided. New/corrected parts are marked to facilitate the assessment of changes. We did our best to fulfil your expectations and we hope that you will be satisfied with our corrections. All the corrections are made in the track changes mode.

This is a review, not a scientific article, therefore aimed at a wide audience of readers, where many of them want to know and understand. The theme of the review should be introduced to the reader in a step-by-step manner, following an almost historical chronology and with logical and clear explanations. We rearranged the text to improve the flow of information and readability

 The authors have created a work that is both compact and verbose, sometimes bordering on chaotic. The authors often report a collage of short sentences taken from many scientific works, which does not allow the reader a continuity of thought. We rearranged the text to improve the flow of information and readability

The authors should divide the review into sub-paragraphs. In terms of the sub-paragraphs, I highly recommend including a dedicated section that focuses on the fascinating topic of stem cells. Specifically, it would be beneficial to explore their unique characteristics as normal cells and delve into their remarkable capacity to undergo a transformation from normal cells to cancerous cells. In fact, the use of stem cells help to repair or replace damaged tissues and organs. They possess the ability to rescue us from diseases that can only be managed with drugs that treat the symptoms. On the other hand, they have also another function in our lives, one that is not as beneficial. They may indeed be the source of some, and perhaps most, cancers. We divided the text into subparagraphs an provided a section concerning stem cells and endometrial stem cells.

Stem cells possess the remarkable ability to renew themselves, migrate freely throughout different tissues, and transform into a wide range of specialized cells. However, stem cells can undergo malignant transformation when these processes are disrupted by gene mutations or certain circumstances. Besides their current focus, the authors should also allocate some attention towards discussing the role of stem cells in telomere maintenance mechanism. The information on stem cells properties as well their role in telomere maintenance mechanism have been provided

The expansion of CSC (cancer stem cells) is commonly linked to the process known as epithelial-mesenchymal transition (EMT), which is considered a general mechanism for the metastasis of cancer cells and even certain virus-infected cells, not only for endometrial cancer. It is important to note that these suggestions are merely recommendations, however, it would also be beneficial for the authors to emphasize also the various functions of stem cells, as well as in relation to the considerations surrounding endometrial cancer. The information of stem cells properties and functions have been provided

Round 2

Reviewer 2 Report

Comments and Suggestions for Authors

I thank the authors for their efforts. The authors have tidied up and summarized the contents of the manuscript. This makes it easily readable. I think that now the review has a flow of information which, step by step, puts even the less experienced reader in a position to learn and understand sequentially. A good job has been done.